# Molecular, Cellular and Functional Changes in the Retinas of Young Adult Mice Lacking the Voltage-Gated K^+^ Channel Subunits Kv8.2 and K2.1

**DOI:** 10.3390/ijms22094877

**Published:** 2021-05-05

**Authors:** Xiaotian Jiang, Rabab Rashwan, Valentina Voigt, Jeanne Nerbonne, David M. Hunt, Livia S. Carvalho

**Affiliations:** 1Centre for Ophthalmology and Vision Science, The University of Western Australia, Perth, WA 6009, Australia; 21636345@student.uwa.edu.au (X.J.); david.hunt@uwa.edu.au (D.M.H.); 2Lions Eye Institute, Nedlands, WA 6009, Australia; rababrashwan@lei.org.au (R.R.); vvoigt@lei.org.au (V.V.); 3Department of Microbiology and Immunology, Faculty of Medicine, Minia University, Minia 61519, Egypt; 4Department of Developmental Biology, Washington University School of Medicine, St. Louis, MO 63110, USA; jnerbonne@wustl.edu

**Keywords:** CDSRR, cone-rod dystrophy, *KCNV2*, *KCNB1*, voltage-gated potassium channels, photoreceptors, retinal degeneration

## Abstract

Cone Dystrophy with Supernormal Rod Response (CDSRR) is a rare autosomal recessive disorder leading to severe visual impairment in humans, but little is known about its unique pathophysiology. We have previously shown that CDSRR is caused by mutations in the *KCNV2* (Potassium Voltage-Gated Channel Modifier Subfamily V Member 2) gene encoding the Kv8.2 subunit, a modulatory subunit of voltage-gated potassium (Kv) channels. In a recent study, we validated a novel mouse model of Kv8.2 deficiency at a late stage of the disease and showed that it replicates the human electroretinogram (ERG) phenotype. In this current study, we focused our investigation on young adult retinas to look for early markers of disease and evaluate their effect on retinal morphology, electrophysiology and immune response in both the Kv8.2 knockout (KO) mouse and in the Kv2.1 KO mouse, the obligate partner of Kv8.2 in functional retinal Kv channels. By evaluating the severity of retinal dystrophy in these KO models, we demonstrated that retinas of Kv KO mice have significantly higher apoptotic cells, a thinner outer nuclear cell layer and increased activated microglia cells in the subretinal space. Our results indicate that in the murine retina, the loss of Kv8.2 subunits contributes to early cellular and physiological changes leading to retinal dysfunction. These results could have potential implications in the early management of CDSRR despite its relatively nonprogressive nature in humans.

## 1. Introduction

First reported in 1983 [1], Cone Dystrophy with Supernormal Rod Response (CDSRR, OMIM 610356) is an inherited autosomal recessive disease that causes lifelong visual loss. A previous study by our group [2] was the first to demonstrate that CDSRR is caused by homozygous or compound heterozygous mutations in the *KCNV2* (Potassium Voltage-Gated Channel Modifier Subfamily V Member 2) gene encoding the voltage-gated K^+^ channel protein subunit Kv8.2. Since then, several other groups have identified more than 40 disease-causing mutations in *KCNV2* [3,4,5,6]. This dystrophy affects the light-sensing rod and cone photoreceptors in the retina and is characterized by reduced visual acuity, photophobia and abnormal color vision that begins in early childhood and deteriorates rapidly by the second decade of life [1,2,7]. Electroretinogram (ERG) testing shows mostly depressed rod (dark-adapted) and cone (light-adapted) photoreceptor activity. However, unlike other cone-rod retinal dystrophies, CDSRR patients present with a supernormal rod ERG response to bright flashes; the rod a-wave response remains reduced and delayed, whereas the rod b-wave becomes supernormal in amplitude under high intensity light stimulus [8,9].

Vertebrate voltage-gated K^+^ channels (Kv) channels are tetramers of four subunits arranged as a ring [10]. Each subunit is composed of two main regions; the pore domain, which allows selective permeability of ion species across the cell membrane, is formed by S5, S6 and a loop that interconnects all subunits; and the voltage-sensor domain, comprising of S1-S4, which undergoes the conformational changes in response to membrane voltage variations [10]. There are around 50 Kv subunits described so far. The Kv8.2 subunit is a member of the KvS (defined as silent or modifier subunits) subfamily that are unable to form homotetrameric channels [7,11]. To form functional channels, they must heterotetramerize with subunits from the Kvα2 family (Kv2.1 or Kv2.2) [9,12,13]. A recent report [14] showed that Kv8.2 is mostly co-expressed with Kv2.1 in the inner segments of rod and cone photoreceptors in mice, macaques and humans, but it is also co-expressed with Kv2.2 in monkey and human cones only. So far, all the reported mutations in *KCNV2* show a fully recessive mode of inheritance, with similar clinical features of the disease [2,8,15,16]. Kv8.2 is the only silent subunit that has thus far been implicated in human disease, but the basis of its unique pathophysiological phenotype and its role in disease development and progression remains unclear, potentially limiting the development of efficient treatment strategies.

In a recent study [17], our group described a novel mouse model of CDSRR that shows a similar ERG phenotype to the one seen in CDSRR patients, including the presence of the supernormal b-wave under a bright light stimulus. This study also showed overall thinning of the retinal outer nuclear layer with significant photoreceptor cell death [17]. In an effort to maximize disease detection, this first study focused on six-month-old animals. However, since visual loss symptoms in CDSRR patients start at a relatively early age, the goal of the present study was to extend our analysis of our mouse model to an earlier disease stage, thereby providing a better evaluation of disease progression. We were also interested in evaluating the potential cellular, molecular and functional changes the absence of Kv subunits can have in the retinas of young adult mice. In this study, we provide a more in-depth morphological and physiological characterization of both Kv8.2 and Kv2.1-deficient mouse models in two-month-old animals, including histological analysis of Kv subunits, photoreceptor markers, outer nuclear layers thickness, quantification of retinal stress and immune profile and functional assessment of visual response. Our data show for the first time that Kv8.2 knockout (KO) retinas still express Kv2.1 subunits, indicating the presence of homotetrameric Kv2.1 channels. We also show that loss of both Kv subunits induces significant photoreceptor death, thinning of the outer nuclear layer, reduction in the photopic visual response, and short and long-term activation of a retinal immune response. These data provide further validation of this KO mouse line as a model for CDSRR, and thereby a useful tool for elucidating the molecular and cellular disease mechanisms. It could also be useful in expanding our understanding of the role of Kv channels in the visual cascade. Finally, it could prove invaluable in assessing and validating novel treatment approaches for CDSRR.

## 2. Results

### 2.1. Localisation of Kv8.2 and Kv2.1 Subunits in WT and KO Retinas

The staining of retinal sections with Kv subunit antibodies showed that both Kv8.2 and Kv2.1 subunits were highly expressed in the WT photoreceptor outer/inner segment (OS/IS) layers as reported previously [14] (Figure 1A). In the KO models, both Kv8.2 and Kv2.1 subunits were absent in the respective KO retinas (Figure 1B,C). Interestingly, expression of Kv2.1 subunits was observed in the Kv8.2 KO retina, indicating the potential presence of homotetrameric Kv2.1 channels in Kv8.2 KO retinas (Figure 1B). However, Kv8.2 subunits were completely absent from Kv2.1 KO retinas (Figure 1C), suggesting that the presence of the Kv2.1 subunit is essential for channel assembly and subunit retention as previously suggested by us [18] and others [11]. Our mRNA expression data (Figure 1D,E) also corroborated Kv subunit expression in the Kv8.2 KO retinas. In the Kv2.1 retinas *Kcnv2* mRNA is present while Kv8.2 protein is absent, indicating that Kv8.2 does not transport to the inner segments when Kv2.1 is absent, as shown in our previous *in vitro* work [18]. The presence of *Kcnb1* mRNA in Kv2.1 KO retinas (Figure 1E) is expected due to how the Kv2.1 KO line has been generated (please see Materials and Methods section). To determine whether Kv8.2 and Kv2.1 are colocalized with mouse cones as well as rods, and if the localization of the homo-tetrameric Kv2.1 subunit in the Kv8.2 KO followed the same pattern, we used a cone-specific marker, cone arrestin, to distinguish between rod and cone outer segments (OS). Figure 2 shows that both Kv8.2 and Kv2.1 subunits are present in both rod and cone photoreceptor IS regions. It also shows that the homo-tetrameric Kv2.1 subunits are localized to the IS region in the Kv8.2 KO retina. Comparing the cone and rod maker staining in the Kv8.2 KO and WT retinas, there is reduced cone arrestin staining in both KO retinas, indicating a possible down-regulation of cone markers and impaired cone outer segments formation (Figure 1B).

### 2.2. Retinal Thickness and Cell Death

The loss of Kv8.2 presents as a progressive cone-rod dysfunction in humans mainly affecting the cone and rod photoreceptors located in the outer nuclear layer (ONL). We, therefore, used paraffin-embedded sections, which better maintain the integrity of the retinal layers, to quantify the thickness of ONL in wild type (WT) and both KO retinas at a relatively early stage of the disease (two months of age). The original thickness of the ONL was calculated in pixels and then converted to μM in ImageJ (Figure 3A,B). A two-way ANOVA with 95% CI was conducted to compare the thickness of ONL in WT, Kv8.2 KO and Kv2.1 KO retinas. The most significant difference was observed in the central retina where both Kv8.2 and Kv2.1 KOs showed a 17% and 30% reduction, respectively, in ONL thickness compared to WT (*p* < 0.002). In the peripheral retina, only Kv2.1 KO showed a significant reduction in ONL thickness at 27% compared to WT (*p* < 0.0001).

In a previous study [17] of the Kv8.2 KO mouse, we showed a significant amount of cell death present in the ONL from one month and at three and six months. However, we did not include Kv2.1 KO retinas, so we have now performed TUNEL (Terminal deoxynucleotidyl transferase dUTP Nick End Labeling) staining on WT, Kv2.1 and Kv8.2 KO retinas at two months of age. Very few TUNEL-positive nuclei were observed in WT and HET mouse retinas, while both Kv2.1 and Kv8.2 KOs had a significant number of TUNEL-positive nuclei in the ONL (Figure 3C). The mean number of TUNEL-positive nuclei for both Kv8.2 KO (*n* = 3, 2950 ± 692) and Kv2.1 KO (*n* = 4, 3432 ± 637) retinas was significantly different from WT (*n* = 4, 264 ± 58.99; WT vs. Kv8.2 KO, *p* = 0.006; WT vs. Kv2.1 KO, *p* = 0.0026). However, the number of TUNEL-positive nuclei from WT retinas did not significantly differ from Kv8.2 HET (*n* = 4, 141 ± 25.3, *p* = 0.1) and Kv2.1 HET (*n* = 3, 227.8 ± 159.6, *p* = 0.8) retinas (data not shown). As expected, the number of TUNEL-positive cells reported here in the two-month-old Kv8.2 KO retinas (Figure 3C) falls in between the number previously reported at one and three months [17], confirming an early-onset progressive cell death. Furthermore, the Kv2.1 KO retina also showed a significant amount of cell death at two months, corroborating the findings of reduced outer nuclear layer thickness shown in Figure 3B, and thereby indicating that the complete absence of Kv channels is detrimental to photoreceptor survival.

### 2.3. Retinal Gliosis and Microglial Activation

As Muller glia and microglia are the predominant glial cell types that maintain the steady-state in the healthy retina, we sought to determine whether loss of Kv8.2 or Kv2.1 subunits was correlated with an altered retinal glia profile as seen in other models of inherited retinal degeneration. We first determined whether the retinas of Kv deficient mice showed signals of retinal stress as indicated by the expression of the glial fibrillary acidic protein (GFAP) in Müller glia cells. GFAP expression in retinal Müller glia is used as an indicator of tissue stress and has been associated with several models of retinal degeneration [19]. Histological analysis and quantification of GFAP protein expression showed an increase in GFAP expression in both Kv2.1 KO and Kv8.2 KO retinas compared to WT (Figure 4A,B). However, when gene expression was assessed, a significant increase was only seen in the Kv2.1 KO retinas (Figure 4C).

Using confocal z-stack images of retinal flatmounts stained with the microglia marker Iba1, we quantified microglia numbers in the inner nuclear layer (INL) and the outer nuclear layer (ONL). In the WT retinas, resident microglia cells exhibited a ramified morphology with a small, round soma, and various branching processes (Figure 4D,E). These inactivated microglia cells were mainly located in the INL. In the Kv8.2 KO retinas, microglia cells assume a reactive state, progressing into phagocytic microglial cells. Under this reactive state, microglia cells proliferate and can assume an amoeboid form (Figure 4E) and migrate into the ONL (Figure 4D) [20].

When quantified, there was a significant 10-fold increase in microglia numbers in the ONL of Kv8.2 KO retinas compared to WT (WT, 3 ± 0.65; Kv8.2 KO, *n* = 3, 31.75 ± 4.53; *p* = 0.0074; Figure 4D); WT 41.25 ± 4.84; Kv8.2 KO, *n* = 3, 76 ± 4.16; *p* = 0.0018), together with a significant increase in microglia numbers in the INL of Kv8.2 KO animals (WT 41.25 ± 4.84; Kv8.2 KO, 76 ± 4.16; *p* = 0.0018). No significant changes compared to WT were found for the Kv2.1 KO retinas. The increased number of microglia in both outer and inner layers of the Kv8.2 KO retina indicates activation and proliferation of microglia cells during disease. The comparison of microglia numbers in the different retinal areas (superior, inferior, temporal and nasal) showed no significant spatial differences.

### 2.4. Long-Term Immune Response

Our data shows a significant increase in microglia numbers in the INL and the presence of activated microglia in the ONL only in the retinas of two-month-old Kv8.2 KO animals. In order to further investigate the effect of disease progression on the localized immune response, we used flow cytometry to characterize and quantify the immune profile of two-month-old WT and Kv8.2 KO retinas via a validated panel of immune markers for retinal samples [21]. Figure 5 shows the different immune cell markers evaluated and total numbers per animal (two eyes pooled). Similar to our histology data, we observed a trend for increased numbers of both microglia and CD45+ cells in Kv8.2 KO, but only CD45+ was statistically significant. However, when microglia were analyzed for CD11c+, an activation marker on microglia normally not expressed in steady-state, we observed a significant increase in the Kv8.2 KO retinas compared to WT. Interestingly, both natural killer cells (NK1.1+) and granulocyte (Ly6G+) were significantly increased in Kv8.2 KO retinas, indicating activation of the innate immune response. However, a slightly increased, but not significant, number of T cells (CD8+ and CD4+) was also observed. To evaluate whether the immune response seen at two months had any long-term effects, we also ran the same panel on six-month-old retinas from WT and Kv8.2 KO mice. As shown in Appendix A, CD45+, NK1.1+, and Ly6G+ cells were all still significantly increased at six months, but activated microglia (CD11c+) were not different from WT.

### 2.5. Visual Function

Despite the supernormal rod response of CDSRR patients, the disorder presents as a cone-rod dystrophy with an early onset of severe photopic vision impairment. Our previous study focused mainly on the scotopic response and characterization of the supernormal response, but also showed that both Kv8.2 and Kv2.1-deficient mice had a significantly impaired photopic flicker response compared to WT. In this study, we examined whether the full flash photopic response is also impaired and how it affects the other visual function measures. As shown in Figure 6A–C, stimulus at lower light intensity was not recordable for both Kv8.2 and Kv2.1 KO retinas (Figure 6B,C). At higher intensities, both the a-wave (Figure 6B) and b-wave (Figure 6C) were recordable for the Kv deficient retinas, but at significantly lower amplitudes than WT.

A reduced optomotor reflex response under photopic conditions was also observed in both models. As shown in Figure 6D, under light-adapted conditions both Kv8.2 and Kv2.1 KO mice showed significantly reduced optomotor responses (clockwise: WT, 17 ± 0.66; Kv8.2 KO, 9.25 ± 0.85; Kv2.1 KO, 9.66 ± 1.20; *p* < 0.0002. Anti-clockwise: WT, 14.33 ± 2.02; Kv8.2 KO, 8 ± 1.42; Kv2.1 KO, 8 ± 1.73; *p* = 0.002).

Our previous study carried out on six-month-old mice [17] showed that the switch to an enhanced positive component of the b-wave in Kv8.2 KO retinas occurred between 0.01 and 0.1 cd.s/m^2^. The data shown in Figure 7 are from two-month-old mice and were obtained from a series of dark-adapted scotopic responses ranging from 0.1 to 25 cd.s/m^2^ (Figure 7B). Figure 7A shows representative scotopic traces from WT, Kv8.2 KO and Kv2.1 KO at 25 cd.s/m^2^ where the enhanced positive component of the b-wave can be seen in both Kv deficient retinas (Figure 7A). The a-wave in both models was significantly reduced compared to WT to a similar extent to thatobserved at six months [17]. However, in contrast to the data from six-month-old animals, the two-month-old Kv2.1 KO retinas also showed an enhanced positive b-wave (Figure 7B). This finding indicates that the Kv2.1 KO retina, which lacks any voltage-gated Kv channels, is also capable, at least at two months, of generating the supernormal b-wave response seen in the Kv8.2 KO CDSRR model.

## 3. Discussion

Since its first report in 2006 [2], several different types of *KCNV2* mutations have now been confirmed to lead to CDSRR. However, the physiological basis between deficient Kv8.2 subunits and the diagnostic supernormal rod ERG response seen in patients is still not well understood. In our previous study, we validated a Kv8.2 KO mouse model of CDSRR [17] and thoroughly characterized its electrophysiological response, showing that voltage-gated K^+^ channels are essential for the generation of normal a-, b- and c-wave responses. We also confirmed that knocking out the *Kcnv2* gene in the mouse also generates a supernormal scotopic b-wave response at higher light intensities, as seen in CDSRR patients.

Clinical studies have shown that CDSRR is a slowly progressing disorder. However, reports of age-related visual acuity changes, ERG responses and macular abnormalities have been variable, and not seen in all patients [3,22,23,24]. Symptoms can start to appear as early as four years of age [8], and retinal structural changes can already be present in patients as young as nine years old [8]. Some observed changes include disruption of the photoreceptor inner segment–outer segment junction and underlying outer segment layer at the fovea, and reduced macular retinal thickness [3,8,22,25]. Our previous study focused on animals at six months of age, which corresponds to a relatively advanced disease stage, but we showed that photoreceptor cell death starts as early as four weeks of age. Therefore, to evaluate whether this early cell death has broader implications in retinal health and function in CDSRR, we examined the cellular and physiological changes at an earlier stage in the disease (two months).

### 3.1. Early Morphological Changes in Potassium Channel Deficient Retinas

Our data corroborate a previous report by Gayet-Primo and colleagues [14] that Kv8.2 is co-expressed with Kv2.1 in the inner segment of mouse cone and rod photoreceptors. However, more importantly, we have now confirmed that in the absence of the Kv8.2 subunit, Kv2.1 subunits can still traffic to the inner segment whereas Kv8.2 subunits do not reach the inner segment when Kv2.1 is absent. This result is in line with our previous in vitro work [18], which showed that the modulatory Kv8.2 subunits need their copartner Kv2.1 to traffic to the cell membrane. As we’ve shown, when Kv2.1 subunits are absent from mouse photoreceptors, Kv8.2 subunits are also missing, indicating, as found in vitro, that Kv8.2 subunits need Kv2.1 to traffic to the inner segments of photoreceptors in vivo.

It seems clear both from our work and from clinical studies that the absence of hetero-tetrameric Kv channels in the retina causes a slowly progressing, but significant, loss of photoreceptor cells from an early stage of the disease. Enhancement of voltage-gated homomeric Kv2.1 channel activity, and subsequent cellular efflux of K^+^ ions, have been shown to accompany apoptotic cell death in cortical neurons [26] and could, therefore, explain the loss of photoreceptors seen in CDSRR. However, the apoptotic surge of Kv2.1 channel activity is only possible via a p38-dependent increase in Kv2.1 phosphorylation [27] and, since retinal Kv2.1 subunits are less phosphorylated than cortical Kv2.1 [14], it could also explain why photoreceptor cell death progresses mildly in CDSRR. However, cellular K+ efflux-mediated apoptosis does not explain the cell death observed in the Kv2.1 KO retina, which was almost double that in Kv8.2 KO (30% vs. 17%, respectively in the central retina). This suggests that the complete absence of voltage-gated K^+^ channels most likely generates a greater ionic imbalance within, and potentially extracellular to, photoreceptors which could then result in the activation of apoptosis independent of K^+^ efflux. Potential differences in photoreceptor cell death mechanisms between Kv8.2 and Kv2.1 KO models are also evident by a discrepancy in the rate of cell death between retinal regions. In the Kv8.2 KO mice, loss of photoreceptors was only significant in the central retina, whereas in the Kv2.1 KO mice there was a similar loss between central and peripheral retina. Despite the lack of a macula, the central loss of photoreceptors in the Kv8.2 KO mouse mirrors the macular thinning reported in CDSRR patients [3]. The comparable rod:cone ratio, and the phagocytic load of the RPE between the peripheral human macular and central mouse retina [28], could account for this similarity and provide further clues towards elucidating cell death mechanisms in CDSRR.

### 3.2. Absence of Kcnv2 Activates an Innate Immune Response in the Retina

The contribution of non-cell-autonomous mechanisms involving inflammation in different types of inherited retinal degeneration (IRD; of which CDSRR is a part) remains unclear, particularly the role of retinal microglia [29,30]. However, there is an increasing appreciation of the non-cell-autonomous contributions that microglia make to the rate and extent of photoreceptor degeneration. One study [31] reported that microglia in mouse models of retinitis pigmentosa (RP) and human RP contribute to photoreceptor degeneration via non-cell-autonomous mechanisms involving primary phagocytosis and inflammatory cytokine production, particularly IL-1b. A continual interaction of microglia and rod photoreceptors, and ultimate phagocytosis of living rods, was also apparent. A following study by the same group showed that the molecular mechanism regulating this microglia-mediated process in IRD models was due to CX3CL1-CX3CR1 signaling, offering potential therapeutic targets [32]. The understanding of non-cell-autonomous mechanisms in IRD could ultimately lead to enhanced therapeutic interventions. Our data show an increase in cell numbers and activated microglia in the Kv8.2 retina at an early stage of the disease. However, this same immune response (microglia activation), was not seen in the Kv2.1 KO retinas, despite these animals showing slightly more retinal degeneration than the Kv8.2 KOs. It is unclear why this is the case, but one possible explanation could be timing, in that microglia involvement in Kv2.1 KO-related degeneration might be happening prior to two months, and this study was too late to capture a proliferative/activated microglia stage. Flow cytometry analysis of the retinas at two months revealed infiltration of natural killer (NK) cells and granulocytes that usually traffic into the eye under inflammatory conditions. NK cells are a central component of the innate immune system and play a critical role in regulating the adaptive immune response. Those cells are known to have an essential role in generating the inflammatory response during uveitis [33,34] and in AMD pathogenesis [35].

Granulocytes produce a wide array of toxic reactive oxygen species and proteases that can further damage ocular cells, while NK cells have cytolytic activity promoting neutrophil infiltration and produce cytokines able to recruit immune cells into the eye [36]. Photoreceptor cells within the retina do not express MHC class I molecules [37], making them vulnerable to NK cell-mediated lysis; therefore, the presence of these cells could represent a threat to the integrity of the retinal environment. We, therefore, believe that the increased presence of both granulocytes and NK cells in our CDSRR model, aligned with our morphological and functional data, suggest that the observed loss of photoreceptor cells could potentially be mediated, or exacerbated, by inflammatory signals.

### 3.3. Potassium Currents and the ERG b-Wave

A study by Gayet-Primo et al. (2018) [14] indicated that photoreceptors might contain both Kv2.1/Kv8.2 heterotetrameric and Kv2.1 homotetrameric channels, with the former responsible for the low voltage-activated *I_kx_* current and the latter for the high voltage-activated current. Since heterotetrameric Kv2.1/Kv8.2 channels are absent from Kv8.2 KO mice, these mice should, therefore, lack the low voltage-activated *I_kx_* current. Our data indicate that Kv2.1 homotetrameric channels are present in the Kv8.2 KO retina, so the high voltage-activated current mediated by homo-tetrameric Kv2.1 channels should be present. In the absence, however, of modulation by the Kv8.2 subunit, these homotetrameric Kv2.1 channels most likely remain closed at low light intensities, and only open in the presence of higher voltages generated by higher light intensities. This might indeed be the case in Kv8.2 KO rod photoreceptors, which show an increase in the amplitude of the supernormal rod response in direct response to increasing light intensities. Cones, however, appear to behave differently. Kv8.2 KO mice show a significantly, and equally, reduced cone flash ERG responses at all light intensities, which is also seen in CDSRR patients [6,7]. A similar reduction in the photopic a-wave is also seen in the Kv2.1 KO mice, so it would appear that the a-wave, which is generated in photoreceptors, requires functional and modulated Kv channels; where they are absent, a smaller non-Kv-dependent response is seen. The photopic b-wave is also depressed in Kv KO mice and does not show the elevated positive b-wave component seen in the scotopic ERGs of these mice.

A reduction in cone visual function was also confirmed in this study by analyzing the optomotor response, showing that a loss in photopic ERG manifests as a loss of visual response. Based on the clinical manifestations and predominantly macular-specific degeneration seen in CDSRR patients [7], it is clear that *KCNV2* mutations affect rods and cones differently. From our data, since the a-wave amplitudes in the Kv8.2 and Kv2.1 KO mice were similarly depressed, it follows that the K^+^ current potentially generated by the homo-tetrameric Kv2.1 channels that are present in the Kv8.2 KO retina, did not contribute to the photoreceptor-driven a-wave response in either rods or cones as proposed in our previous study [17]. Only modulated Kv channels were, therefore, involved in this response, and it would appear that the full a-wave amplitude was achieved by a modulated Kv channel component and a smaller non-Kv component that remained in the KO mice.

A supernormal b-wave in CDSRR patients is diagnostic in many cases and is replicated in the Kv8.2 KO mice by an enhanced positive component of the b-wave [17]. This enhanced response was also seen in the two-month-old Kv2.1 KO mice used in this study, which contrasts with its absence in Kv2.1 KO mice at six months [17]. Its presence in two-month-old Kv2.0 KO mice implies that this enhanced response may not be due to the presence of homo-tetrameric Kv2.1 channel-derived high voltage-activated currents, because they are absent in Kv2.1 KO mice. In their study, Gayet-Primo and colleagues [14] hypothesized that unmodulated homotetrameric Kv2.1 channels depolarize the rods’ resting potential, thereby affecting the voltage-gated calcium current. This would have the effect of shutting down neurotransmitter release under bright light stimuli to generate a supernormal postreceptoral b-wave response. If the presence of unmodulated Kv2.1 channels (as seen in the Kv8.2 KO) has the same effect on rod depolarization as a complete lack of Kv channels (as seen in the Kv2.1 KO), this would explain the supernormal b-wave in young Kv2.1 KO retinas but not its absence at six months of age. The latter absence may arise, therefore, from the detrimental effects of the continued lack of functional Kv channels as the retina develops.

The b-wave response combines changes in the membrane potential of bipolar and Muller glia cells induced by photoreceptor activation via glutamate release that depolarizes bipolar cells, and an increase in extracellular K^+^ that depolarizes Müller glial cells, respectively [38]. We therefore propose a modified hypothesis for the supernormal b-wave, i.e., that a complete absence of Kv channels or presence of the unmodulated Kv2.1 homotetrameric channels creates an imbalance in extracellular K^+^ which alters the depolarization rate of rod bipolar and/or Muller glial cells. This extracellular K^+^ imbalance would be transient in Kv2.1 KO retinas and only present at early ages. With time, and as a result of an absent outward rectifying K^+^ current in photoreceptors, the supernormal b-wave response in Kv2.1 KO retinas disappears. In the Kv8.2 KO retina, however, the presence of homotetrameric Kv2.1 channels would still produce a K^+^ current, albeit only at high voltages (e.g., high light stimuli), but continuing over time. This is consistent with the current paradigm that both bipolar and Muller glia cells contribute to the b-wave, and the increase in extracellular K^+^ picked up by Muller glia comes most likely from the depolarized bipolar cells [39,40]. The presence of inward-rectifying K^+^ channels in Muller glial cells [41] and outward rectifying K^+^ channels in bipolar cells [42] could potentially support this hypothesis. Furthermore, Muller glia’s role in K^+^ siphoning from the extracellular space into the vitreous is well known [43,44], and our GFAP protein expression data indicate that in both Kv8.2 KO and Kv2.1 KO retinas, Muller glia are under stress. Interestingly, Muller glia stress seems to be more pronounced in Kv2.1 KO retinas (mRNA and protein level), which aligns with the higher number of cell deaths observed in this line. However, it might also suggest that the complete lack of Kv2.1 subunits has a higher impact on extracellular K^+^ levels, potentially overworking Muller glia cell’s K^+^ siphoning capacity. Although the importance of K^+^ to the b-wave is now widely accepted, no studies to date have yet been able to determine the molecular mechanism of how Muller glia, bipolar cells and K^+^ interact to give rise to the ERG b-wave. The data presented here, and our previously published work on the Kv8.2 and Kv2.1 KO models [17], indicate that these mice models could be excellent tools to help further our understanding of vision physiology.

## 4. Materials and Methods

### 4.1. Animals

All mice were group-housed in a climate-controlled facility on a 12-h light/dark cycle with food and water ad libitum. All experiments described below were performed in animals at two months of age unless stated otherwise.

A full description of the Kv2.1 knockout (KO) line can be found in [45]. Our colony was established with animals obtained from Professor Jeanne Nerbonne’s laboratory at the Department of Developmental Biology, Washington University School of Medicine, St. Louis, MO, USA. Heterozygous carriers (Kv2.1^+/−^) were intercrossed to generate Kv2.1^+/+^, Kv2.1^+/−^ and Kv2.1^−/−^ littermates for experiments. The Kv2.1^−/−^ line is referred to as Kv2.1 KO in this manuscript. The Kv8.2 KO line was generated at the Wellcome Trust Sanger Institute Hinxton, UK. A full description of how this line was generated can be found in [17]. Homozygote carriers (Kv8.2^−/−^) were intercrossed to generate homozygote animals for experiments and are referred to as Kv8.2 KO in this manuscript.

### 4.2. Immunohistochemistry and Imaging

Enucleated eyes from Kv2.1 KO, Kv8.2 KO and wild type (WT) mice were fixed for 1 h at room temperature in 4% paraformaldehyde (PFA, Electron Microscopy Sciences, Inc., Hatfield, PA, USA) in 1X phosphate-buffered saline (PBS, Sigma-Aldrich, North Ryde, NSW, Australia). After fixation, the cornea, lens and sclera were removed, and eyecups placed back in 4% PFA for an additional 30 min at room temperature. Eyecups were then cryoprotected in a 20% sucrose (Sigma-Aldrich, North Ryde, NSW, Australia) solution overnight at 4 °C, embedded in optimal cutting temperature (O.C.T compound; Tissue-Tek, Sakura Finetek Inc. Torrance, CA, USA) media and sectioned at 14 um using a Leica (CM 3050S) cryostat. Sectioning was done on the sagittal plane (nasal to temporal direction) so that sections are on the dorsal-ventral axis.

For staining of retinal sections, nonspecific binding sites were blocked for 1 h at room temperature in block solution containing 10% normal goat serum, 0.5% Triton X-100, (Sigma-Aldrich, St. Louis, MO, USA) 1% BSA (Sigma-Aldrich, St. Louis, MO, USA) in 1X PBS. Primary antibodies were diluted in block solutions and applied to sections overnight at room temperature (Appendix A). After washing with 1X PBS the next day, sections were incubated in secondary antibody diluted in block solution (1:500) for 2 h at room temperature. Secondary antibodies were raised in goat and conjugated to AlexaFluor-488 (Abcam, Cambridge, MA, USA), or -568 (Abcam, Cambridge, MA, USA). Slides were then incubated in DAPI (4′,6-diamidino-2-phenylindole, 0.5 μg/mL in 1X PBS) for 10 min to label cell nuclei and mounted using Dako mounting medium (Agilent Technologies, Santa Clara, CA, USA). For flat-mount Iba1 staining, eyes were enucleated and fixed in 4% PFA overnight at 4 °C. Flat-mount retinas were marked at the superior part of the cornea and dissected into four quadrants indicating superior, nasal, inferior and temporal parts of the retina. After fixation in 4% PFA at 4 °C for an extra 1 h, retinas were incubated in 10% normal goat serum, 3% Triton X-100, 1% BSA in 1X PBS for 1 h at room temperature. Primary antibody incubation was done overnight at room temperature (Appendix A). The retinas were then washed with 1X PBS and labeled with secondary antibody also overnight at 4 °C at a 1:500 dilution in block solution. All images were acquired on a Nikon A1Si confocal microscope located at the UWA Harry Perkins Centre for Microscopy, Characterization and Analysis (CMCA, Nedlands, WA, Australia) and were taken from the dorsal and/or ventral areas.

#### Quantification of Immunohistochemistry Images

Quantification of retinal outer nuclear layer thickness was done on formalin-fixed paraffin sections stained with hematoxylin and eosin (H&E). Briefly, eyes were fixed in Davidson fixative for 24 h at room temperature. They were then incubated for 24 h in 10% formalin and embedded in paraffin. Sections were cut as described above (sagittal plane) at 10 µM and stained with H&E. For quantification of the ONL thickness, central and periphery images were taken as following. Central images were taken on each side of the optic nerve (dorsal and ventral) at around 10° (central) and 80° (periphery) from the optic nerve. Three sections were imaged per animal, with two central (one dorsal and one ventral) and two periphery (one dorsal and one ventral) images taken (total of 12 images per eye).

For quantification of Iba-1 staining, confocal *z* stacks of vertical sections of retina labelled for Iba-1 and DAPI were acquired on a Nikon A1Si confocal microscope. Central images from the dorsal, ventral, nasal and temporal areas (20× magnification, four images per eye) were taken, and quantified using an automated count in Fiji ImageJ (National Institutes of Health, USA). In some cases, adjustments to image brightness and contrast were made with ImageJ (Fiji) and Photoshop (Adobe, San Jose, CA, USA).

GFAP fluorescence intensity was determined using NIS Elements (v5.3, Tokyo, Japan) for the three different mice lines (*n* = 3 mice/line). For each retinal section, three different images were selected from different areas (dorsal and ventral) in the retina to be quantified. The desired area was outlined between the outer limiting membrane and the inner limiting membrane of the retina using the ROI tool (Region of interest). The thresholding values were determined using control tissue sections (WT) and applied to all images for analysis. The percentage of the fluorescent area was calculated for each image and averaged across each line. No difference in fluorescence intensity was observed between dorsal and ventral areas.

### 4.3. Apoptosis

Terminal deoxynucleotidyl transferase (TdT)-mediated dUTP nick end labeling (TUNEL) was performed using the ApopTag^®^ Red In situ Apoptosis Detection kit (MilliporeSigma, Burlington, MA, USA). Staining followed the manufacturer’s protocol and TUNEL-positive cells were counted on an Olympus Fluorescent Microscope BX60.

### 4.4. RNA Extraction and Real-Time Quantitative Polymerase Chain Reaction (qPCR)

Total RNA was extracted from whole retinas using TriReagent (Sigma-Aldrich) as per the manufacturer’s instructions. Reverse transcription was performed using QuantiTect Reverse Transcription Kit (Qiagen, Hilden, Germany) as per manufacturer’s instructions. Quantitative Real-Time PCR (qPCR) was performed on a Bio-Rad CFX Connect Real-Time System using Taqman Fast Advanced master mix (Thermo Fisher Scientific, Waltham, MA, USA) with the following gene assays: *Gfap* (Mm01253033_m1), *Kcnv2* (Mm00807577_m1); *Kcnb1* (Mm00492791_m1) and *Gapdh*, (Mm99999915_g1). Gene expression was normalized to *Gapdh* and relative expression calculated using the ΔΔCt method.

### 4.5. Electroretinogram (ERG Recordings)

Retinal function was evaluated via full-field flash scotopic and photopic electroretinogram (ERG) measurements using the HMsERG system (OcuScience LLC, Rolla, MO, USA). Mice were dark-adapted overnight and handled subsequently only under dim red light. Mice were anesthetized with 2% isoflurane in 100% oxygen at a flow rate of 1 L/min and pupils dilated by applying 1% tropicamide (MYDRIACYL^®^; Alcon, Geneva, Switzerland) to the surface of the cornea. A drop of 2% hypromellose (GONIOVISC, HUB Pharmaceuticals, LLC, Rancho Cucamonga, CA, USA) solution was also applied to the cornea to keep it moist throughout the recordings and prior to placing the electrodes on the eye. Sedated animals were placed on a heating pad kept at 37 °C, and a stainless-steel ground electrode was placed sub-dermally above the base of the tail, and reference electrodes were placed sub-dermally in each cheek along the jawline in an anterior direction. The eye electrodes combined a silver thread with a contact lens and were placed on top of the cornea for each eye. The mice were then placed under the Ganzfeld dome to ensure a uniform illumination stimulus was presented.

Scotopic and photopic recordings followed previously described protocols [46] and consisted of the following. For scotopic recordings, animals were dark-adapted for 12 h, and single-flash recordings were obtained through the presentation of 1 ms flashes (four repeats) with the following intensities (all in cd.s/m^2^): 0.1, 0.3, 1, 3, 10, 25. The time interval between consecutive repeats was 10 s at 0.10 Hz. A recovery time of 60 sec was present between the different intensities. For photopic recordings, animals were light-adapted for 10 min at 30 cd.s m^−2^, followed by a series of flashes (32 repeats) on the 30 cd.s m^−2^ background at 2 Hz at the following intensities (all in cd.s m^−2^): 1, 3, 10, 25. The time interval between consecutive flashes was 0.5 s. Before analysis of b and a-waves, low-pass 150 Hz filtering was applied. Data were analyzed using the ERGView Software (4.380R; OcuScience LLC) and Excel (Microsoft^®^).

### 4.6. Optomotor Response

An optokinetic drum was used to measure head tracking responses to moving stimuli, as described previously [47]. Mice (*n =* 3–4 per genotype) were placed in a no reflective transparent container positioned on the central pedestal of the drum. Each stimulus consisted of a square-wave grating with vertical black and white stripes of equal width placed against the inside wall of the drum. Movement speed was set at two revolutions per minute, the optimal rate for eliciting an optokinetic response in mice [48]. After 5 min acclimatization in the drum, the stripes were rotated clockwise for 2 min, then anticlockwise for 2 min, with a 30 s pause for the change of direction. Tests were performed under mesopic light range (1000 lux), and a video was recorded. Head-tracking movements in response to the moving stimulus were only considered if they followed the direction and angular speed of the rotating stripes (1 revolution/min) and lasted for more than one second. The number of tracking movements during the 2 min test in each direction were averaged for each mouse.

### 4.7. Flow Cytometry

Single-cell suspensions of retinal tissue were prepared at the indicated collection age. Briefly, eyes were dissected to separate the anterior from the posterior segment. The posterior segment was dissected to separate the retina from the choroid/sclera. Retinas from both globes were pooled for each mouse before the tissue was manually homogenized and digested in a mixture of 10 µg/mL Liberase (Roche, Basel, Switzerland , Cat No #05401119001) and 10 µg/mL DNAse I (Sigma, USA) in 1X PBS for 40 min at 37 °C. The resulting single-cell preparations were stained with antibodies specific for CD45 (30F11), CD11b (M1/70), CD3 (145-2C11), CD4 (RM4-5), CD8 (53-6.7), NKp1.1 (PK136), CD11c (HL3), CD19 (6D5), CD64 (X54-5/7.1), F/480 (BM8), MHC-II (M5/114), Ly6C (AL.21), Ly6G (1AB). Antibodies were obtained from BD Biosciences (San Jose, CA, USA), BioLegend (San Diego, CA, USA), or eBioscience (San Diego, CA, USA). Fixable viability stain 620 (BD Biosciences) was used for live/dead discrimination. Samples were analyzed using an LSRFortessa X-20 instrument (BD Biosciences). The gating strategies used to identify immune cell populations in the retina are shown in Appendix A. All data analysis was performed using the FlowJo software package (FlowJo, LLC, Ashland, OR, USA).

## Figures and Tables

**Figure 1 ijms-22-04877-f001:**
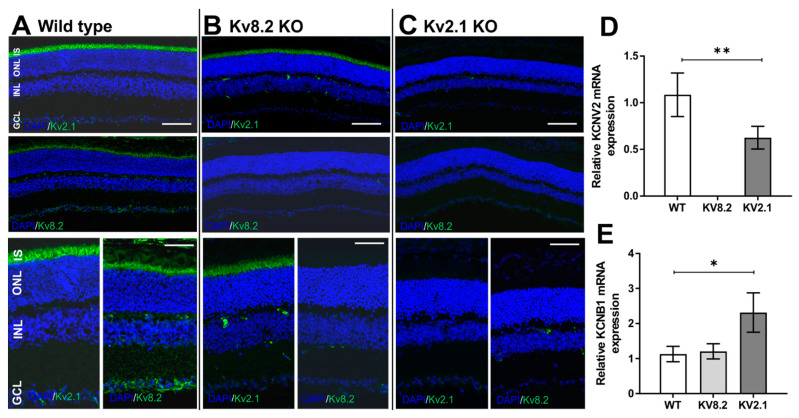
Retinal localization and expression of the voltage-gated K^+^ channel protein subunits Kv8.2 and Kv2.1 in wild type (WT), Kv8.2 knockout (KO) and Kv2.1 KO. (**A**–**C**) Representative confocal images of the inner segment (IS) area, outer nuclear layer (ONL), inner nuclear layer (INL) and ganglion cell layer (GCL) showing expression of Kv8.2 and Kv2.1 mouse subunits in wild type (**A**), Kv8.2 KO (**B**) and Kv2.1 KO (**C**) retinas at 20× (top panels) and 40× (bottom panels) magnification. Scale bars = 20 μM (top panels) and 50 μM (bottom panels). (**D**,**E**) Relative mRNA expression of *Kcnv2* and *Kcnb1* genes in all three lines. Results are presented as mean +/− SD from *n* = 3 (WT and Kv8.2 KO) and *n* = 4 (Kv2.1 KO), and *p* values were obtained through one-way ANOVA and Dunnett’s multiple comparison test post hoc with * *p* = 0.0094 and ** *p* = 0.0001.

**Figure 2 ijms-22-04877-f002:**
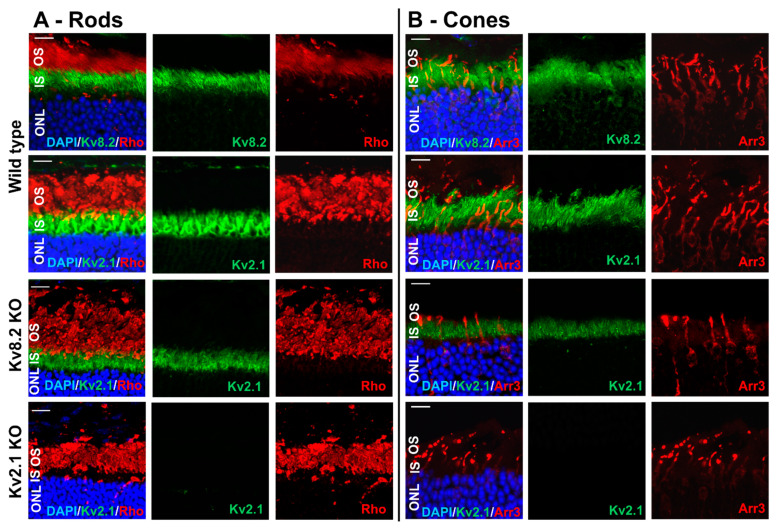
Representative confocal images of wild type and knockout (KO) retinas from central (10° from optic nerve) dorsal-ventral areas showing signal overlap of (**A**) rhodopsin (Rho, red) or (**B**) cone arrestin (Arr3, red) with both Kv subunits in wild type and Kv2.1 subunit in Kv8.2 KO retinas (green). No anti-Kv8.2 and anti-Kv2.1 staining was observed in the Kv2.1 KO retinas. ONL, outer nuclear layer; IS, inner segment; INL, inner nuclear layer; OS, outer segment. Scale bar = 20 μM.

**Figure 3 ijms-22-04877-f003:**
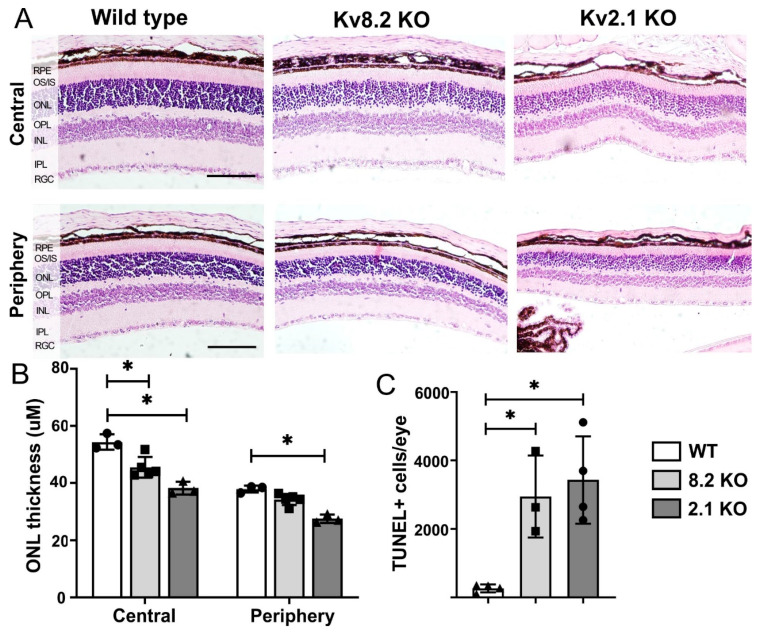
Retinal thickness and cell death in wild type (WT) and knockout (KO) retinas. (**A**) Representative histological retinal sections from the dorsal-ventral region of KO models and WT showing the different retinal layers. RPE, retinal pigment epithelium; OS/IS, outer segment/inner segment; ONL, outer nuclear layer; OPL, outer plexiform layer; INL, inner nuclear layer; IPL, inner plexiform layer; RGC, retinal ganglion cells. Scale bar = 100 μM. (**B**) Quantification of outer nuclear layer (ONL) thickness in the different mouse models in central (10° from optic nerve) and peripheral (80° from optic nerve) areas of the retina. No differences were observed between dorsal or ventral areas. The results are presented as mean +/− SEM, with statistical analysis by two-way ANOVA and Sidak’s multiple comparison test post hoc with * *p* < 0.002. (**C**) Quantification of cell death in the ONL of WT, Kv8.2 KO and Kv2.1 KO retinas via TUNEL (Terminal deoxynucleotidyl transferase dUTP Nick End Labeling) staining. Results are presented as mean +/− SD from *n* = 3–5, and *p* values were obtained through one-way ANOVA and Tukey’s multiple comparison test post hoc with * *p* < 0.006.

**Figure 4 ijms-22-04877-f004:**
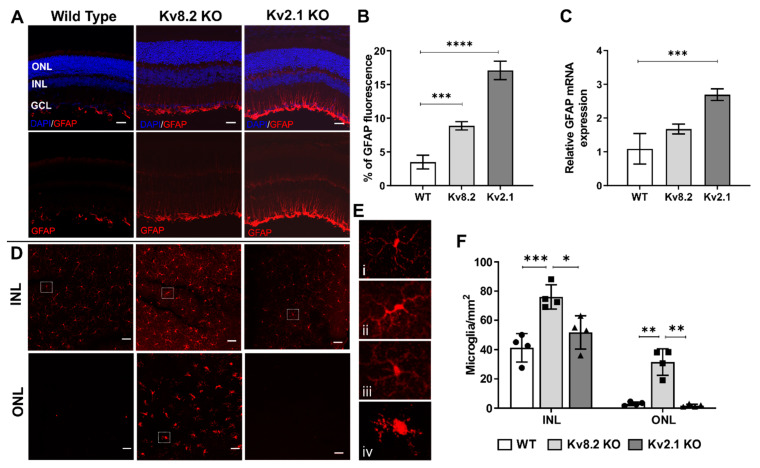
Glia activation in Kv deficient retinas. (**A**) Glial fibrillary acidic protein (GFAP) retinal expression in wild type, Kv8.2 KO and Kv2.1 KO animals. Upper panels show nuclear stain (DAPI, blue) and GFAP (red) co-localisation. Boomt panels show GFAP (red) expression only. Scale bar = 50 μM (**B**) Fluorescent quantification of GFAP protein expression in the three mouse lines showing significantly higher expression in both Kv KO lines compared to WT. (**C**) Quantification of *Gfap* gene expression via qPCR showing a significant increase in expression in Kv2.1 KO retinas compared to wild type (*n* = 3 for each genotype). (**D**) Confocal images of retinal flatmounts labeled with microglia marker Iba-1 (red) taken at the inner nuclear layer (INL) and outer nuclear layer (ONL) regions to show proliferation and migration of activated microglia in the Kv8.2 KO retina. Scale bar = 50 μM. (**E**) Higher magnification images of individual microglia cells from WT (i), Kv8.2 KO INL (ii) and ONL (iii), and Kv2.1 KO INL (iv) showing the morphological differences in cell body shape. (**F**) Quantification of microglia numbers in the INL and ONL in the wild type, Kv8.2 KO and Kv2.1 KO retinas. Results are presented as mean +/− SD from *n* = 3, and *p* values were obtained through two-way ANOVA and Tukey’s multiple comparison test post hoc with * *p* < 0.02, ** *p* < 0.004, *** *p* < 0.0006, **** *p* < 0.0001.

**Figure 5 ijms-22-04877-f005:**
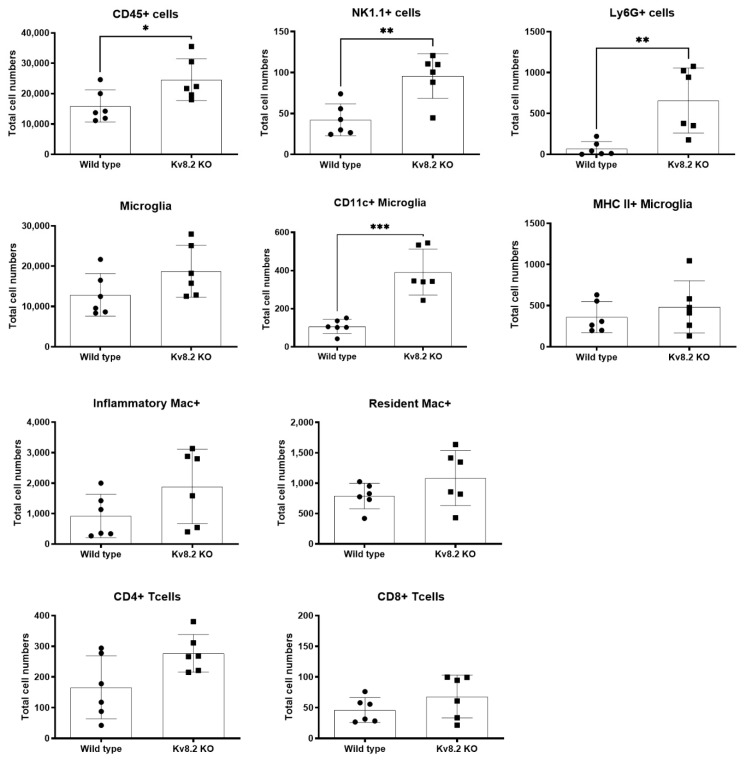
Characterization of immune cells present in the retina of wild type (WT) and Kv8.2 knockout (KO) mice. Panels show quantification of different types of immune cells in two-month-old WT and Kv8.2 KO retinas. Results are presented as mean +/− SEM from *n* = 6. *p* values were obtained through unpaired *t*-test with Welch’s correction, * *p* = 00286, ** *p* < 0.0029 and *** *p* = 0.0002.

**Figure 6 ijms-22-04877-f006:**
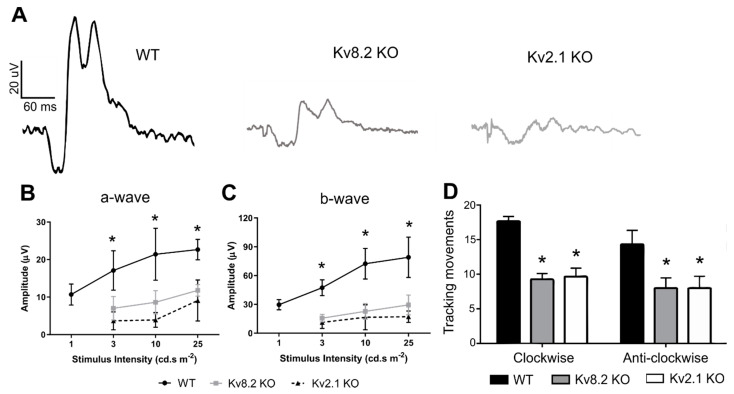
Photopic visual response from Kv deficient retinas. (**A**) Representative light-adapted photopic full flash ERG traces from wild type (WT), Kv8.2 knockout (KO) and Kv2.1 KO animals taken at 25 cd.s/m^2^. (**B**) Quantification of photopic a-wave in all three genotypes at different stimulus intensities. * denote significance between the Kv lines and WT with *p* < 0.0005. No statistical difference was observed between the Kv8.2 and Kv2.1 KO lines. (**C**) Quantification of photopic b-wave in all three genotypes at different stimulus intensities. * denote significance between the Kv lines and WT with *p* < 0.0005. No statistical difference was observed between the Kv8.2 and Kv2.1 KO lines. (**D**) Light-adapted photopic optomotor response from all three genotypes. *Y*-axis shows tracking movements per 2 min. * *p* < 0.002 compared to WT. Results for (**B**–**D**) are presented as mean +/− SD from *n* = 3–4 animals (6 eyes per genotype). Significance was obtained through a two-way ANOVA and Tukey’s multiple comparison test post hoc.

**Figure 7 ijms-22-04877-f007:**
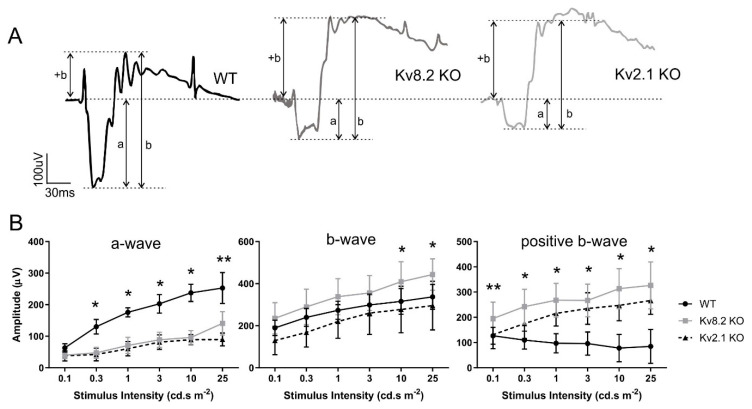
Scotopic ERG response. (**A**) Representative traces of dark-adapted full flash scotopic responses at 25 cd.s/m^2^. Traces show the distinctive supernormal b-wave (+b) component of the ERG trace in both Kv8.2 and Kv2.1 KO animals. Amplitude length of a-wave (a) and full b-wave (b) are also shown. (**B**) Quantification of the dark-adapted scotopic a-wave, b-wave and positive b-wave components at different light intensities for WT, Kv8.2 KO and Kv2.1 KO animals. Results are presented as mean +/− SD from *n* = 3–4 animals (six eyes per genotype); significance was obtained through a two-way ANOVA and Tukey’s multiple comparison test post hoc. a-wave: * *p* < 0.0001 at all intensities for both Kv lines compared to WT; ** *p* = 0.0085 between Kv8.2 and Kv2.1 KO lines. b-wave: * *p* < 0.01 between Kv8.2 and Kv2.1 KO lines only. Not significant compared to WT. Positive b-wave: ** *p* = 0.0025 only between WT and Kv8.2 only. * *p* = 0.004 (0.3 cd.s m^−2^) and *p* < 0.0006 (1–25 cd.s m^−2^) between Kv lines and WT. Not significant between Kv lines.

## Data Availability

Not applicable.

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
