# Peer review of "Molecular, Cellular and Functional Changes in the Retinas of Young Adult Mice Lacking the Voltage-Gated K+ Channel Subunits Kv8.2 and K2.1"

_ijms, 2021, doi:10.3390/ijms22094877_

Round 1

Reviewer 1 Report

Overall, this manuscript provides some tantalizing data. But the paper lacks a clear research question and results-based justification for each subsequent outcome measure. Thus, the manuscript lacks a cohesive plot. 

1) Title: The authors characterize the effects of Kv8.2-/- and Kv2.1-/- in the retina of 2 month old mice. Because this analysis was carried out using only 2 month old mice, I believe the use of the word "development" may misinform the readers. In my mind a "development" signifies a longitudinal or cross-sectional study. I would suggest "... in the young adult mouse retina". 

2) Abstract: I have identified several instances of grammar/usage errors.  Second sentence is very long; please split the sentence into two sentences. 

"...in a previous study we had validated..." delete "had"

"...we gave demonstrated that the retinas in the Kv KO mice..." replace with retinas of Kv KO mice. 

Please replaced all instances of KO and HET with appropriate -/- and +/-. 

The last sentence of the abstract seems far reaching. Please tone down the tenor. 

3) Introduction: The first paragraph lacks a clearly defined research question. 

Line 3 remove "has".

Several instances of citation errors i.e., [6,[7] instead of [6, 7].

Usage error: "...S5, S6 and a loop that interconnects of all".

Is hetero-tetramerize a valid word?

Please define "silent subunit". 

Last sentence of second paragraph: An argument from ignorance is weak. 

Typo: "focussed".

Usage: i.e., "six months old" change to six month old. .

Usage: photoreceptors death change to photoreceptor death. 

Third sentence from the end is a run on sentence. 

Last sentence, again is far reaching. Please tone down tenor. 

The last few sentences of the introduction should include a restatement of your research question, how you approached the question, results, and how your findings fit into and extend the literature. 

4) Results. 

Figure 1: For general interest, please include vertical sections of whole retina with higher resolution of ONL and IS.

To support "homo-tetramer" conclusion, the same sections should be labeled with both Kv8.2 and 2.1. 

Figure 2: Authors state "co-expression of rhosopsin and cone arrestin". Co-expression is not quantified. Instead use the term signal overlap. 

Figure 3: 

Please include brief description of methods used in the results section. For example how were cells counted and how were retinas oriented. 

Please use the greek mu (micro) symbol instead of "u". 

Authors state "ONL was calculated in pixels and then converted to a percentage". However, Figure 3B measurement unit is micrometers. Please be consistent.

Figure 4: Authors state "In the Kv8.2 KO retinas, microglia assume a reactive state...". Please show high resolution examples and morphological analysis for this or remove. 

Please provide quantification of GFAP. 

Figure 4 legend Kv8.2 instead of "Kv82".

Figure 6: Please add labels to identify a- and b-wave. 

Subject number (n) is quite low for all outcome measurements. 

Pg 8 second paragraph first sentence please correct "00.1". 

5) Discussion:

Pg. 10, first paragraph, second sentence: "active homo-tetrameric Kv2.1 channels". The results do not confirm activity. 

6) Methods:

1M PBS, should this instead be 1X PBS?

Usage: Second paragraph: "superior of the corneal". 

Reviewer 2 Report

In this paper, Jiang et al present the morphological and electrophysiological characterization of two novel mouse models of Cone Dystrophy with Supernormal Rod Response (CDSRR) at 2 months of age. While the authors have already recently published characterization of these animal models at later ages (6 months), this paper provides further insight into the timing of phenotype development and early retinal changes occurring in these models. The paper is overall clear and well organized. However, there are some aspects which need to be revised in order to increase the clarity of the manuscript, as detailed below:

  • Figure 2A: Rho staining in Kv8.2 KO mice appears to be not optimal. Indeed, outer segments (OS) appear to be shortened and not clearly elongated. Please either comment this apparent disorganization of rod OS in Kv8.2 KO mice or, if the picture was not representative of an actual phenotype observed in this model, replace with a more representative image. Also, in the legend to Figure 2, panels A (rho) and B (arrestin) are mislabeled, as follows “…co-expression of (B) rhodopsin (Rho, red) or (C) cone arrestin”.
  • Given the not uniform distribution of short wavelength-sensitive and middle/long wavelength-sensitive opsins (S- and M-opsin) in the mouse retina, the authors should better describe in which retinal location each of the histological analysis presented has been performed (dorsal-ventral and nasal-temporal). Accordingly, for data in Fig.2B and 3C staining with markers of the different cone types is suggested.
  • Figure 3: As data in panel A show different rates of retinal thinning in central vs peripheral retina, the TUNEL data in panel C should be split to separately express data coming from the two different retinal locations, too. Also, the sentence starting in the last line of page 4 states: “The mean number of TUNEL-positive nuclei for both Kv8.2 KO (n= 3, 4275 ± 692) and Kv2.1 KO (n=4, 3432 ± 637) retinas was significantly different to WT”. The mean values included in parenthesis do not seem to correspond to the numbers included in the graph in Fig. 3C; please either clarify the discrepancy or correct.
  • As the authors propose that microglial activation contributes to the worsening of the disease, they should discuss the finding that Kv2.1 KO retinas show more profound retinal degeneration compared to Kv2.8 (Fig.3) despite no significant changes in microglia numbers compared to WT (Fig. 4).
  • Page 10, first paragraph: “Kv2.1 subunits are still capable of trafficking to the inner segment and forming active homo-tetrameric Kv2.1 channels”. How can the author define whether these assembled channels are active? Please discuss more.
  • Please provide further details on how all the analysis on retinal sections/flatmounts have been performed: how many pictures for each eye have been analysed? Where in the retina they were acquired (see comment above)? How many regions have been counted in each image?
  • Text needs a careful revision as there are some typos, including sentences apparently lacking references and listing “(ref)”.

Reviewer 3 Report

General:

The paper is well written; the flow reads well, especially the discussion is very thorough and the study is indeed important regarding the understanding of early cellular and physiological markers of CDSRR with photoreceptor degeneration, and thereby in the evaluation of disease progression. The paper is on the light side regarding amount of presented data.

The authors should cite other groups that identified a linked from mutations in the KCNV2 gene to the disease (for example in the introduction, 4.th line. Suggestions: Wissinger et al., iovs 2008; Zellinger et al., Ophthalmology, 2013; Abdelkader et al., Doc Ophthalmol., 2020).

The knockouts are not conditional. How does the knocking out of Kv2.1 (or Kv8.2) affect the animals in general?

Minor concern is that the language has some has errors/typos.

Comments for sections:

  • Figure 1: When knocking out Kv2.1 Kv8.2 expression is also abolished. The opposite is not case indicative of Kv2.1 being required for subunit retention and assembly. This was already proposed byCzirják et al., 2007 and Smith et al., 2012. The authors could cite these works (Results 2.1 line 6-7).
  • The authors write, in the paragraph starting immediately after the figure legend of Fig.3, page 4: ”In a previous study [15] of the Kv8.2 KO mouse, we showed a significant amount of cell death present in the ONL from 1 month and at 3 and 6 months. However, we had not included Kv2.1 KO retinas, so we have now performed TUNEL staining on WT, Kv2.1 and Kv8.2 KO retinas at 2 months of age.” Then on page 5, line 5-6 the authors write: “These results are in line with our previous study, which quantified the number of TUNEL positive cells in 1, 3 and 6 months old retinas [15].”? This should either be corrected or rephrased, so that it is clear that the new findings are in line with what was previously found for the Kv8.2 KOs and WTs.
  • The GFAP expression pattern in Kv2.1 KO should be discussed further (Fig.4). Even though the qPCR revealed a non-significant increase in GFAP compared to WT clear changes are seen. Maybe not on mRNA level, but histomorphologically. The Müller cells distribution (or the number of MCs) is obviously altered compared to WT. Did the authors investigate protein expression by means of Western Blot? If not this is a key experiment.
  • Just a suggestion: The authors could label the graphs in Fig.6B+C, like in Fig.7, to make it easier to follow the data (‘a-wave’ and ‘b-wave’). This is minor.

Round 2

Reviewer 1 Report

The authors have responded to all of my previous comments with appropriate revision or rebuttal. 

Reviewer 3 Report

My only (very) minor comment is that some parts of the text should be superscripted, words should be line divided correct and the remaining few typos should be corrected. Otherwise the manuscript is acceptable as is.